# Transcriptome Analysis of Developmental Gene Expression in *Thesium chinense* Turcz

**DOI:** 10.3390/plants14162549

**Published:** 2025-08-16

**Authors:** Sijia Liang, Qiongqiong Wang, Qin Han, Xinmin Zhang, Yiyuan Liu, Miaosheng Chen, Chengcai Zhang, Zhaoyang Wang, Junxiao Li, Di Yu, Hao Zhan, Yubin Zhang, Zhongping Xu

**Affiliations:** 1Academy of Industry Innovation and Development, Huanghuai University, Zhumadian 463000, China; qchzxm@126.com (X.Z.); 18739605666@163.com (Y.L.); 18739656616@163.com (M.C.); zhaoyang332277@163.com (Z.W.); lijunxiao2004@163.com (J.L.); y1595d@163.com (D.Y.); 18007033845@163.com (H.Z.); 13809532498@163.com (Y.Z.); 2College of Plant Protection, Henan Agricultural University, Zhengzhou 450046, China; qqwang@henau.edu.cn; 3National Key Laboratory of Crop Genetic Improvement, Huazhong Agricultural University, Wuhan 430070, China; 4Agricultural Technology Promotion and Plant Protection Quarantine Station, Zhumadian 463000, China; hanqin-zmd@163.com; 5State Key Laboratory for Quality Ensurance and Sustainable Use of Dao-Di Herbs, National Resource Center for Chinese Materia Medica, China Academy of Chinese Medical Sciences, Beijing 100700, China; zhangchengcai2021@126.com

**Keywords:** *T. chinense*, transcriptome, DEGs, network, flavonoid biosynthesis

## Abstract

*Thesium chinense* Turcz. (*T. chinense*), a perennial herb in the Santalaceae family, exhibits potent antibacterial and anti-inflammatory properties. Transcriptome sequencing was performed on one- and two-year-old *T. chinense* plants across seedling, flowering, and fruiting stages (all sampled from the same location) using the illumina NovaSeq 6000 platform. A total of 58,706 unigenes were identified, including 1656 transcription factors (TFs). Further analysis classified these TFs into seven functional categories, enabling the reconstruction of a representative TF regulatory network. Differential expression analysis revealed that the number of differentially expressed genes (DEGs) ranged from 2000 to 5000 during different developmental stages in first-year plants, while varying between 1000 and 2000 in second-year plants. Comparative analysis of DEGs between one- and two-year-old plants showed that they were primarily associated with sesquiterpene, triterpene, and terpene skeleton biosynthesis, as well as other metabolic pathways. Additionally, analysis of key genes involved in flavonoid biosynthesis—the major bioactive compounds in *T. chinense*—revealed their predominant accumulation during the first year of growth. This study provides valuable insights into the developmental biology of *T. chinense* and establishes a foundation for future research on flavonoid biosynthesis pathway genes and their therapeutic applications.

## 1. Introduction

*Thesium chinense* Turcz. (*T. chinense*), a perennial hemiparasitic herb belonging to the Santalaceae family, is widely used in traditional Chinese medicine (TCM) for treating inflammatory and infectious diseases [1]. This medicinal plant possesses multiple therapeutic properties, including heat-clearing, detoxification, and cough/phlegm relief, making it particularly effective against inflammatory conditions such as mastitis, pneumonia, tonsillitis, laryngopharyngitis, and upper respiratory tract infections. Its remarkable efficacy has earned it the nickname “plant antibiotic” [2]. *T. chinense* exhibits a biennial growth habit as a hemiparasitic plant: First-year plants display weak growth, minimal branching, and an extended growth cycle, persisting predominantly in a seedling stage. Conversely, second-year plants develop from dormant root neck buds, exhibiting clustered growth with a relatively shorter growth cycle [3]. Current harvesting relies primarily on wild populations owing to the lack of effective large-scale cultivation methods. Consequently, this species faces critical challenges: (1) sparse natural population distribution, (2) small individual plant size with low biomass yield, and (3) rapid decline of wild resources due to overharvesting and habitat degradation. Therefore, enhancing both the yield and bioactive compound content in *T. chinense* represents an urgent research priority in medicinal plant studies.

Phytochemical studies have identified kaempferol and its glycosides—classified as flavonoid compounds—as the primary bioactive constituents of *T. chinense*. Notably, kaempferol content and total flavonoid levels serve as critical quality assessment biomarkers for this species [4]. Studies have shown that the total flavonoid content in *T. chinense* ranges from 3.21% to 3.77%, with an average of 3.38%, while the kaempferol content varies between 0.9 and 1.1 mg·g^−1^. Although the primary chemical constituents of *T. chinense* from different populations are similar, their contents exhibit variations, some of which may be statistically significant [2]. Owing to their potent antibacterial and anti-inflammatory properties, *T. chinense* extracts are extensively incorporated into clinical formulations (e.g., tablets and capsules) to meet substantial market demands [5]. Significantly, these extracts inhibit both replication and cellular entry of severe acute respiratory syndrome coronavirus 2 (SARS-CoV-2, the COVID-19 causative agent) while exerting potent anti-inflammatory effects [6]. However, current research on *T. chinense,* both domestically and internationally, remains predominantly focused on geographical distribution patterns [7,8], fundamental biological characteristics, phytochemical composition [9], pharmacological mechanisms [10], and quality assessment methodologies [11]. Consequently, the biosynthetic pathways, transport mechanisms, regulatory networks, and molecular foundations of these active compounds are poorly characterized [12]. This knowledge gap critically constrains the targeted development of bioactive constituents and the implementation of precision breeding strategies. Medicinal plant-derived flavonoids exhibit remarkable therapeutic potential through multifaceted pharmacological activities, including antioxidant, anti-inflammatory, antimicrobial, antitumor, and cardioprotective effects [13]. Structurally diverse subclasses (e.g., flavonoid glycosides, isoflavones, anthocyanins) exert anti-inflammatory actions via dual modulation of inflammatory mediator secretion and inflammatory cell activation [14]. Specific flavonoids, like resveratrol and quercetin, demonstrate pronounced anticancer properties by simultaneously inhibiting neoplastic cell proliferation and inducing apoptotic pathways, highlighting their clinical relevance in oncology [15]. Nevertheless, the intricate structural diversity among flavonoid subclasses necessitates further investigation into their precise pharmacological mechanisms and structure–activity relationships [16,17]. While flavonoid biosynthetic pathways and associated genetic determinants are well characterized in model plants, the metabolic networks governing secondary metabolite production in *T. chinense* remain uncharted territory.

To bridge this knowledge gap, the present study aims to (1) decipher the stage-specific transcriptional regulation of flavonoid biosynthesis across key developmental phases (seedling, flowering, fruiting) in one- and two-year-old *T. chinense* plants; (2) construct co-expression regulatory networks centered on transcription factors to elucidate hierarchical control of terpenoid and flavonoid pathways; and (3) identify core genes governing the accumulation of bioactive flavonoids, particularly during the first growth year where maximal phytochemical yield occurs. These objectives will establish a molecular framework for the targeted enhancement of bioactive compound production in this medicinally significant species.

This study employed high-throughput RNA sequencing to analyze transcriptomic dynamics in *T. chinense* across biennial growth cycles and developmental stages, with the primary objective of constructing a transcription factor-centered co-expression network to decode stage-specific regulatory mechanisms governing flavonoid biosynthesis. Through comprehensive bioinformatic analysis, we identified and functionally annotated key regulatory genes associated with the flavonoid biosynthesis pathway—the primary source of bioactive compounds in *T. chinense*. These findings provide novel molecular insights into the mechanistic basis of phytochemical accumulation and quality formation in this medicinal plant.

## 2. Results

### 2.1. Transcriptome Sequencing of Different Stages of T. chinense

To elucidate the growth dynamics of *T. chinense*, we conducted transcriptome sequencing and profiling across six key developmental stages spanning its biennial cycle: first-year seedling (Tc1_S), flowering (Tc1_F), and fruiting (Tc1_Fr) stages, followed by second-year seedling (Tc2_S), flowering (Tc2_F), and fruiting (Tc2_Fr) stages using the Illumina NovaSeq 6000 platform (Appendix A). From these samples, 18 cDNA libraries were constructed, yielding an average of 21 million high-quality reads per library, with a total data volume of 113.37 Gb, providing a robust dataset for comprehensive transcriptomic analysis (Table 1).

### 2.2. Assembly and Annotation of T. chinense Transcripts Across Different Developmental Stages

Using RNA-seq data from 18 cDNA libraries, we performed de *novo* transcriptome assembly and annotation of *T. chinense* via Trinity, generating 161,311 transcripts and 58,706 unigenes. The transcript length distribution was relatively uniform across size categories (300–500 bp, 500–1000 bp, 1–2 kb, and >2 kb), each containing > 30,000 transcripts (Figure 1; Appendix A). Assembly metrics indicated an average transcript length of 1805 bp with an N50 of 2878 bp. For unigenes, most (<1 kb) exhibited an average length of 1306 bp and an N50 of 2316 bp (Figure 1A; Appendix A). BUSCO (Benchmarking Universal Single-Copy Orthologs) assessment confirmed high assembly quality, revealing 58.2% complete single-copy and 1.8% complete duplicated unigenes, totaling 60.0% complete BUSCO genes (Figure 1B). Functional annotation across five major databases assigned putative functions: Nr (NCBI non-redundant protein, 58.95%), KOG (eukaryotic orthologs, 21.13%), GO (gene ontology, 46.41%), Pfam (protein families, 46.42%), and NT (nucleotide, 54.07%) (Appendix A). Notably, 9144 unigenes (15.58%) received annotations in all five databases (Figure 1C). From the assembled transcripts, 45,070 protein sequences were predicted (Appendix A). Protein length distribution analysis showed 84.33% spanned 200–400 amino acids (AA), while 98.72% were <1000 AA (Figure 1D). Additionally, we identified 1656 transcription factors (TFs) representing 57 plant TF families (Appendix A), providing key insights into regulatory networks during *T. chinense* development.

### 2.3. Stage-Specific Gene Expression Patterns in T. chinense Development

Using the assembled transcriptome as a reference, clean reads from each sample were aligned, with mapping rates ranging from 73% to 79% (Figure 2A). Comparative analysis of FPKM (Fragments Per Kilobase per Million mapped reads) values revealed distinct gene expression distributions across samples (Figure 2B). Correlation analysis confirmed strong consistency among biological replicates at all time points (Figure 2C), validating data reliability. These results collectively demonstrate the robustness and biological relevance of the identified gene expression profiles.

Differential expression analysis revealed dynamic gene expression patterns across developmental periods. In the Tc1 phase, 22,214 genes showed constitutive expression, whereas stage-specific genes were identified as follows: 6173 in Tc1_S (seedling), 1934 in Tc1_F (flowering), and 3758 in Tc1_Fr (fruiting). Similarly, the Tc2 phase contained 26,800 commonly expressed genes, with stage-specific genes distributed as: 2544 in Tc2_S, 4966 in Tc2_F, and 4851 in Tc2_Fr (Figure 2D). This indicates that although most genes maintain multi-stage expression, distinct subsets are uniquely expressed at specific developmental points. Further analysis of nine stage-pair comparisons (Tc1_S vs. Tc1_F, Tc1_F vs. Tc1_Fr, Tc1_S vs. Tc1_Fr, Tc2_S vs. Tc2_F, Tc2_F vs. Tc2_Fr, Tc2_S vs. Tc2_Fr, Tc1_S vs. Tc2_S, Tc1_F vs. Tc2_F, Tc1_Fr vs. Tc2_Fr) revealed contrasting interannual patterns: Tc1 exhibited 2000–5000 DEGs (|Log2FC| > 1, *p*-value < 0.05) with predominant upregulation, while Tc2 showed 1000–2000 DEGs with predominant downregulation (Figure 2E,F and Appendix A). This differential regulation likely reflects developmental programming differences linked to extended inter-phase durations.

Temporal clustering analysis identified twelve distinct TF expression patterns (Appendix A). Notably, Cluster 6 TFs exhibited gradual upregulation across developmental stages (Figure 3A), with stage-specific expression variations validated by heatmap analysis (Figure 3B). These findings highlight the temporal regulatory specialization of TFs during *T. chinense* development, where specific TF families display distinct stage-dependent expression profiles.

### 2.4. Expression Patterns of Key TFs in T. chinense

To elucidate TFs involved in gene regulatory networks (GRNs), we used GENIE3 to identify 48 conserved plant TF families exhibiting significant expression variations across developmental stages, classified into seven subfamilies based on their expression patterns (Figure 4A). Subfamily 1 (e.g., HSF, C2C2-YABBY) showed low expression in Tc1 but marked upregulation in Tc2_Fr; Subfamily 2 (AP2/ERF-ERF, MYB) displayed sustained high expression from early Tc1 through the Tc2 stage; Subfamily 3 (C2C2-CO-like, NAC, MADS-MIKC) maintained consistently high expression in Tc2; Subfamily 4 (CSD) peaked in early Tc2; Subfamily 5 (EIL, WRKY) exhibited consistently high expression in Tc2; Subfamily 6 (FAR1) was preferentially expressed during Tc1_F/Fr; and Subfamily 7 (SBP, MYB-related) achieved maximal expression in Tc1_Fr (Figure 4B, Appendix A). Collectively, this systematic TF profiling enabled GRN reconstruction, revealing stage-specific regulatory mechanisms mediated by key TF families during *T. chinense* development.

### 2.5. Co-Expression Analysis Reveals Developmentally Relevant Functional Networks in T. chinense

We investigated co-expression networks across *T. chinense* growth stages by analyzing gene expression profiles. Soft-threshold power calculations based on scale-free topology criteria evaluated gene-gene correlations, with all expression correlations visualized (Appendix A). Weighted gene co-expression network analysis (WGCNA) identified 20 distinct modules, among which the ME turquoise module exhibited a robust association with the Tc1_S stage (Figure 5A). Network visualization revealed that 99% of nodes in this module were differentially expressed across developmental timepoints, and *PSK*, *ZFP*, *OATP,* and *WRKY* were identified to be at key positions in the module (Figure 5B), indicating core regulatory roles in growth. Functional annotation showed ME turquoise module enrichment in key KEGG pathways, including protein kinases, plant-pathogen interaction, MAPK signaling, plant hormone signal transduction, glycosyltransferases, and circadian rhythm, and GO terms, including response to chemical stimuli, multicellular organismal processes, defense responses, and hormone signaling (Figure 5C). Temporal clustering of multicellular developmental processes identified progressively upregulated gene clusters, notably *Cluster-2623.15518* and *Cluster-2623.16869,* showing consistent upregulation (Figure 5D). These results elucidate stage-specific regulatory networks and identify key candidate genes coordinately regulating *T. chinense* development.

### 2.6. Flavonoid and Terpenoid Metabolic Dynamics During T. chinense Development

To delineate functional roles of DEGs, we performed temporal clustering analysis of their expression profiles. Dynamic expression fluctuations indicated stage-specific functional specialization of gene sets (Figure 6A). KEGG annotation of stage-specific DEGs revealed more pronounced functional shifts in Tc1 compared with Tc2, particularly during early Tc1 (Tc1_S vs. Tc1_F), encompassing pathways such as amino acid synthesis/metabolism, photosynthesis, phenylpropanoid biosynthesis, monoterpenoid biosynthesis, MAPK signaling, limonene/pinene degradation, flavonoid biosynthesis, and fatty acid metabolism (Figure 6B). Critically, while monoterpenoid biosynthesis and limonene/pinene degradation persisted throughout development, flavonoid biosynthesis was predominant in early stages. Flavonoid biosynthetic genes exhibited progressive accumulation initiating in early Tc1 (Tc1_S vs. Tc1_F) and continuing throughout growth. Specifically, genes such as *TT5* (*Chalcone-flavanone isomerase*), *CYP98A3* (*cytochrome P450*), and *CCOAMT* (*caffeoyl-CoA 3-O-methyltransferase*) exhibited high expression levels in the early stage of Tc1. Genes including *F3H* (*flavanone 3-hydroxylase*), *TT4* (*Chalcone and stilbene synthase*), *FLS1* (*flavonol synthase 1*), and *CHIL* (*Chalcone-flavanone isomerase*) reached maximum expression during the Tc1_F stage. *HCT* (*hydroxycinnamoyl-CoA shikimate/quinate hydroxycinnamoyl transferase*) and *2OG* (*2-oxoglutarate and Fe (II)-dependent oxygenase*) genes showed high expression levels during the Tc2 stage. The *C4H* (c*innamate-4-hydroxylase*) gene had its highest expression level in Tc1_S, while the *DFR* (*dihydroflavonol 4-reductase*) gene exhibited significant changes during the Tc1_Fr period. These data indicate that flavonoid synthesis during *T. chinense* growth is primarily concentrated in the Tc1 early stage (Figure 6C and Appendix A).

Temporal clustering of DEGs (Figure 7A and Appendix A) identified Cluster 6 genes with progressive upregulation during development (Figure 7B). These genes were enriched in KEGG pathways, including chaperones/folding catalysts and monoterpenoid biosynthesis, and GO terms, including carotenoid/tetraterpenoid metabolism, photosynthesis, and tetraterpenoid biosynthesis (Figure 7C,D). These results indicate a strong association of Cluster 6 with flavonoid and terpenoid synthesis. Collectively, temporally coordinated flavonoid-terpenoid accumulation, mediated by stage-specific gene regulation, drives developmental adaptation in *T. chinense*.

### 2.7. qRT-PCR Validation Confirms Transcriptomic Data Reliability in T. chinense

To validate the reliability of our transcriptomic findings, we performed qRT-PCR analysis on six key flavonoid biosynthetic genes (*TT5*, *CYP98A3*, *TT4*, *CHIL*, *HCT*, and *2OG*) across eighteen samples representing six developmental stages. Quantitative results demonstrated highly consistent expression trends between the transcriptomic and qRT-PCR data (Figure 8).

## 3. Discussion

Despite *T. chinense*’s limited biomass yield and sparse natural distribution, its broad-spectrum phytotherapeutic properties confer significant medicinal value. To elucidate developmental gene expression dynamics, we conducted transcriptome sequencing at key developmental transitions. Systematic analysis revealed stage-specific regulatory patterns: Tc1 exhibited 2000–5000 DEGs with upregulated dominance, whereas Tc2 showed 1000–2000 predominantly downregulated DEGs, indicating intensified metabolic regulation during early growth. Strikingly, interannual comparisons demonstrated disproportionate downregulation in Tc2, likely reflecting developmental prioritization of specific pathways. Functional enrichment analysis linked these DEGs to critical processes including phenylpropanoid biosynthesis, monoterpenoid production, MAPK signaling, limonene/pinene catabolism, and flavonoid biosynthesis. These findings substantially advance understanding of *T. chinense* molecular physiology and establish a genetic framework for targeted enhancement of flavonoid/terpenoid biosynthesis to boost yield and diversity of pharmacologically active compounds.

### 3.1. Transcriptome Assembly Quality and Research Limitations in T. chinense

The absence of a *T. chinense* reference genome has hindered current research primarily to resource characterization, phytochemical identification, and gene expression profiling [8,10,18]. Although our de novo transcriptome assembly achieved moderate completeness (60% BUSCOs; Figure 1B)—a value that requires cautious interpretation given BUSCO’s primary design for genome assessment—this outcome reflects technical and biological constraints. Specifically: (1) BUSCO scores in transcriptome assemblies are influenced by sequencing depth, tissue specificity, and evolutionary distance from model species. (2) *T. chinense*’s uncharacterized genome and hemiparasitic biology limit ortholog detection. Despite these constraints, 60% completeness remains biologically meaningful for a non-model plant with no prior genomic resources, supported by robust functional annotation (Appendix A). In addition, the reproducibility of transcriptome analysis due to sequencing depth, genomic DNA residue and template concentration, and the biological differences of independent samples and sampling location during the sampling process are also problems that need to be improved in future research (Figure 2C). Notably, TF annotation identified 57 conserved families (Appendix A), including key developmental regulators (C2H2 [19], AP2/ERF [20], bHLH [21], bZIP [22], GRAS [23], and MYB [24]). Critical knowledge gaps persist, including unidentified “hub genes” governing Tc1/Tc2 transitions and unverified mechanisms for predominant Tc2 downregulation (potentially reflecting resource reallocation to reproduction). Comparative analysis underscores significant biological resource gaps relative to established medicinal species (*Catharanthus roseus* [25], *Cannabis sativa* [26], *Lonicera* [27], *Panax ginseng* [28]). These limitations collectively emphasize that a high-quality reference genome is imperative to (1) resolve transcriptome assembly ambiguities affecting BUSCO accuracy; (2) elucidate hub gene networks defining developmental transitions; and (3) decipher functional drivers of Tc2 transcriptional downregulation.

### 3.2. Multiple Biological Pathways Coordinate and Regulate the Growth and Development of T. chinense

Transcriptomic analysis demonstrates that *T. chinense* development involves coordinated activation of multiple genetic pathways. Co-expression network analysis specifically implicates multicellular organismal processes as critical regulators (Figure 5C,D). During its biennial cycle, temporal gene accumulation patterns were observed for monoterpenoid biosynthesis [29,30], MAPK signaling [31], limonene/pinene degradation [32], and ascorbate metabolism and α-linolenic acid pathways. Notably, flavonoid biosynthesis—a primary source of bioactive constituents in *T. chinense*—exhibited stage-specific regulation peaking during first-year growth (Figure 5B), a pattern quantitatively validated by qRT-PCR (Figure 6). These flavonoids possess multifaceted therapeutic activities, including antioxidant [15,33], anti-inflammatory [14], anticancer [34], cardioprotective [35], antimicrobial [36], and antiviral [37] properties. Collectively, the stage-resolved coordination of these evolutionarily conserved pathways—particularly flavonoid biosynthesis—mechanistically substantiates *T. chinense*’s efficacy in traditional Chinese medicine.

### 3.3. TFs Regulate the Expression of Genes Related to Flavonoid Synthesis in T. chinense

Through integrated co-expression network analysis, we identified key TFs in *T. chinense* and their target gene networks. Notably, the *FAR1* TF was predicted to regulate *CYP98A3*—a pivotal flavonoid biosynthetic gene (Appendix A), consistent with prior reports of *FAR1*-*CYP98A3* co-expression [38] and *FAR1*’s developmental regulatory roles [39]. This finding raises a fundamental question: Could targeted manipulation of *FAR1* or associated TFs enhance flavonoid production? In silico evidence suggests that engineering such TF-gene interactions—via transgenic approaches or selective breeding—holds promise for boosting bioactive compound yields. Therefore, this study elucidates spatiotemporal expression patterns and regulatory mechanisms of flavonoid biosynthetic genes, providing a molecular framework for (1) strategically optimizing flavonoid production through TF-mediated gene regulation; (2) resource utilization via precision breeding; and (3) molecular authentication of *T. chinense* germplasm.

## 4. Materials and Methods

### 4.1. Plant Materials

Samples of *T. chinense* were collected from the experimental base of Huanghuai University in Zhumadian City, Henan Province, China (33°0′40″ N, 114°0′4″ E) during the growing seasons of 2021 and 2022. At this facility, *T. chinense* primarily parasitizes *Dendranthema indicum* (wild chrysanthemum) and was cultivated using standardized field protocols. All plants were grown under natural field conditions, with uniform soil management, irrigation regimes, and fertilization programs implemented to ensure healthy growth across all developmental stages. For both 2021 and 2022, stems and leaves of robust one-year-old and two-year-old *T. chinense* plants were selected at the seedling (S), flowering (F), and fruiting (Fr) stages. Post-collection, tissues were aseptically rinsed with distilled water to remove surface contaminants, blot-dried using sterile absorbent paper, immediately wrapped in pre-chilled aluminum foil, and flash-frozen in liquid nitrogen to preserve RNA integrity. Three biological replicates (independent plants) per developmental stage per year were processed under identical conditions to ensure experimental reproducibility.

### 4.2. RNA Extraction, Library Preparation, and Sequencing

Total RNA was extracted from all samples using the RNAprep Pure Polysaccharide and Polyphenol Plant Total RNA Extraction Kit (Centrifugal Column Type, Tiangen, Beijing, China, Cat# DP441), following the manufacturer’s protocol. The total amounts and integrity of RNA were assessed using the RNA Nano 6000 Assay Kit on the Bioanalyzer 2100 system (Agilent Technologies, Santa Clara, CA, USA). For library construction, the NEBNext^®^ Ultra^TM^ II RNA Library Prep Kit for Illumina^®^ (NEB, Ipswich, MA, USA, Cat# E7775L) was used according to the manufacturer’s instructions. The prepared libraries were sequenced on the Illumina Novaseq 6000 platform (Illumina, San Diego, CA, USA). Sequencing parameters and technology details (e.g., depth, coverage, quality filtering) were provided by Novogene Co., Ltd. (Beijing, China). Accurate qRT-PCR-based quantification required effective library concentrations >2 nM. Raw data quality control was performed using fastp (v0.19.7) with the following parameters: -g -q 5 -u 50 -n 15 -l 150.

### 4.3. RNA-Seq Data Analysis

After the clean reads were obtained, they were assembled into contigs using Trinity software (v2.6.6, default parameters) for subsequent reference sequence analysis [40]. Transcripts were clustered into groups according to shared reads using Corset based on the Trinity assembly, and new “genes” were established by combining transcript expression levels across different samples [41]. The completeness of the transcriptome assembly was assessed using BUSCO [42]. Subsequently, gene function annotation was performed using the Nr (NCBI non-redundant protein sequences), KOG (Eukaryotic Orthologous Groups), GO (Gene Ontology), Pfam (Protein family), and NT (Nucleotide sequence) databases. Clean reads from each sample were mapped to the Trinity-assembled transcriptome reference (Ref) using RSEM software (v1.3.3), and reads with alignment quality values below 10 were filtered out [43].

### 4.4. CDS Prediction

CDS prediction was performed using a two-step process. First, genes were aligned against the NR and SwissProt protein databases, following a priority order recommended by previous studies [44]. If a match was found, the open reading frame (ORF) information was extracted from the alignment results, and the coding region sequence was translated into an amino acid sequence using the standard codon table. Second, for sequences that either did not match the NR or SwissProt databases or failed to yield predicted results from the alignment, TransDecoder (v3.0.1, https://github.com/TransDecoder/TransDecoder accessed on 26 February 2023) software was used to predict their ORFs. This step produced both the nucleotide sequence and the amino acid sequence for these coding genes. Finally, results from the two steps were merged.

### 4.5. TFs Regulation of Gene Networks

Plant TF prediction was conducted using the iTAK (v1.2) software, which was based on the principle of utilizing transcription factor family classifications and rules defined in the database, and then identifying TF through hmmscan [45,46]. Next, the R package GENIE3 (v1.10.0) was employed to construct a transcription factor-centered gene regulatory network (GRN) [47]. Transcription factors were considered as potential regulators of gene expression profiles in the network. A random forest method with 1000 trees was used as the threshold, with a threshold value set at 0.001.

### 4.6. Enrichment Analysis

Differential expression analysis was conducted using DESeq2 (v1.20.0) [48]. The resulting *p*-values were adjusted with the Benjamini and Hochberg method to control the false discovery rate, and significant differential expression was determined by setting a threshold of p.adj < 0.05 and |log2(fold change)| > 1. Subsequently, GO functional enrichment analysis and KEGG pathway enrichment analysis were performed on the differentially expressed gene sets using GOseq (1.10.0) and KOBAS (v2.0.12) software [49,50].

### 4.7. WGCNA Analysis

Weighted Gene Co-expression Network Analysis (WGCNA), a systems biology approach for describing gene association patterns across samples, was implemented using the R package WGCNA, which provided various functions for weighted correlation analyses [51]. The input data file comprised FPKM values for each sample, and co-expression networks were visualized and enhanced using Cytoscape software (v3.9.1) [52].

### 4.8. qRT-PCR

The RNA from all samples was extracted separately, and then RNA was reverse-transcribed into cDNA using the HiScript II 1st Strand cDNA Synthesis Kit (+gDNA wiper) (R212) from Nanjing Novogene Bioinformatics Technology Co., Ltd. Specific primers for the selected genes were designed with SnapGene software (v6.0.2), with 18S rRNA as the internal reference (gene and primer details are mentioned in the attachment). Quantitative real-time PCR (qPCR) was performed using the ChamQ SYBR Color qPCR Master Mix (Q411) kit from Nanjing Novogene Bioinformatics Technology Co., Ltd., with each reaction conducted in quadruplicate in a 15 μL system. The relative expression levels of the selected genes were analyzed using the 2^−ΔΔCT^ method. Primer sequences for all relevant genes are listed in Appendix A.

### 4.9. Other Related Methods

Graphs were commonly generated using R scripts. The ggplot2 package (v3.3.5) was used to create visualizations such as boxplots and line plots [53], and the ComplexHeatmap package (v2.4.3) was employed for the generation of heatmaps [54].

## 5. Conclusions

This study establishes the comprehensive transcriptomic framework for *T. chinense*, revealing stage-specific flavonoid biosynthesis peaks during the first-year flowering stage through temporally regulated expression of key genes and TFs. We further identified distinct regulatory dynamics between growth years—with first-year plants exhibiting enriched DEGs linked to sesquiterpene/triterpene pathways—and constructed a TF-centered co-expression network that deciphers hierarchical control of phytochemical accumulation. Critically, these insights provide a foundation for precision breeding targeting flavonoid-enriched cultivars and metabolic engineering of “plant antibiotic” compounds.

## Figures and Tables

**Figure 1 plants-14-02549-f001:**
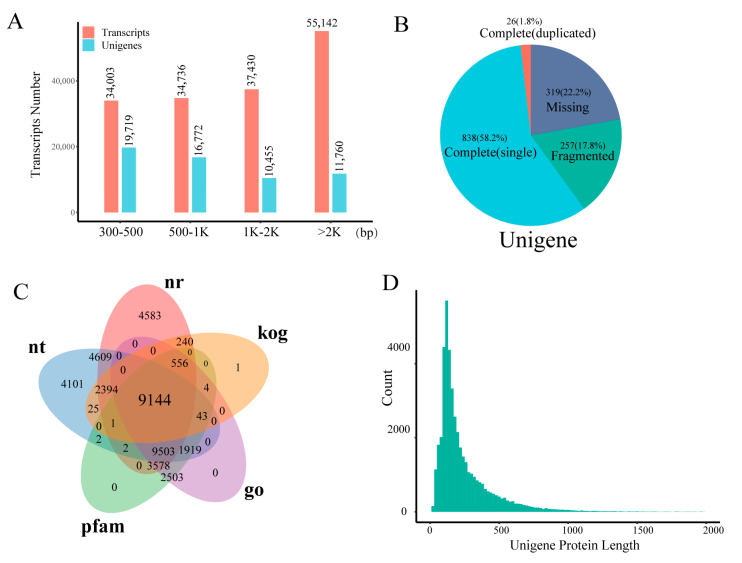
*T. chinense* transcriptome assembly quality. (**A**) Transcriptome assembly transcript and unigene distribution. (**B**) Completeness of unigene detection. (**C**) The number of annotations for the unigenes in five databases, with each color representing a database and overlapping colors indicating overlapping gene counts shared between databases. (**D**) The CDS predictions of unigenes determine the distribution of proteins.

**Figure 2 plants-14-02549-f002:**
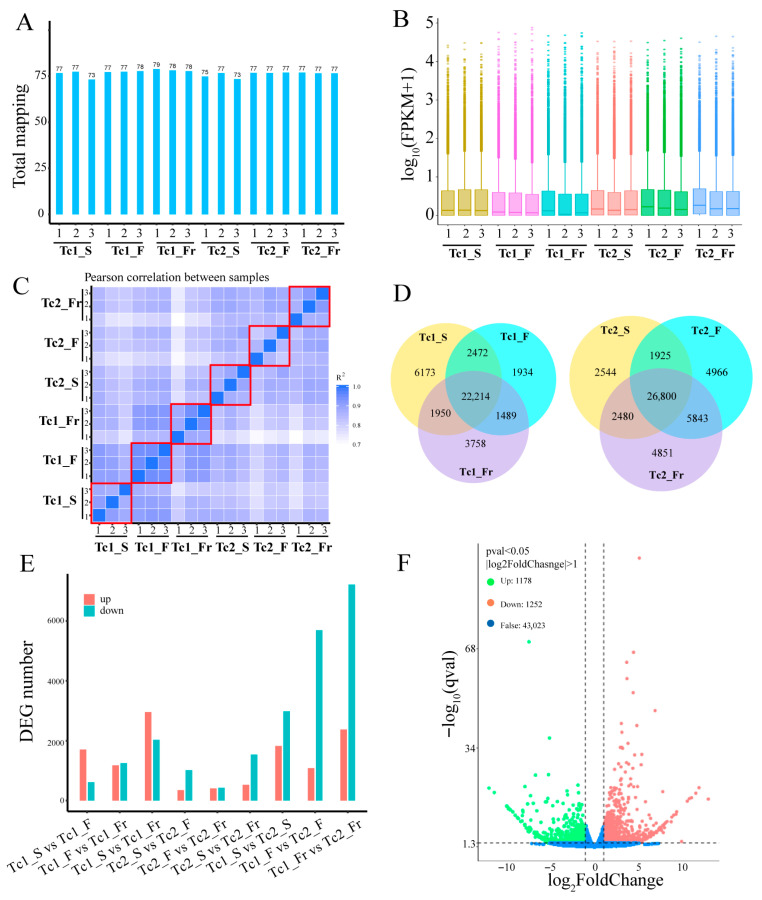
Transcriptomic landscapes and differential expression signatures. (**A**) Sample mapping ratio: the X-axis represents samples, and the Y-axis represents mapping rate. (**B**) Sample expression level: the X-axis represents samples, and the Y-axis represents the log10 value of FPKM+1. The same color indicates replicates. (**C**) The correlation analysis between samples was performed using the FPKM values matrix. Three biological replicates of each sample are represented within the red boxes. (**D**) The Venn diagram shows the number of expressed genes at each time period, with each color representing a different time point. (**E**) The number of differentially expressed genes at different time points. (**F**) Volcano plot of Tc1_F vs. Tc1_Fr.

**Figure 3 plants-14-02549-f003:**
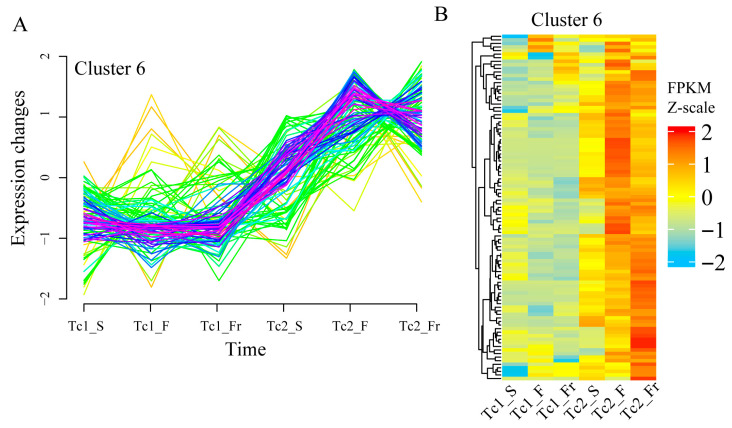
Trends in TF expression. (**A**) Trends in the expression level of transcription factor Cluster 6. (**B**) Transcription factor expression heatmap of Cluster 6, with colors representing the normalized FPKM values; each row represents a gene.

**Figure 4 plants-14-02549-f004:**
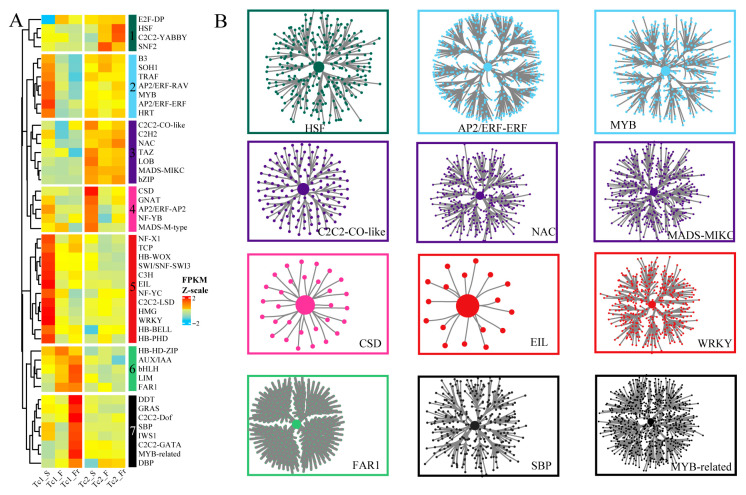
Identification of TF families and regulatory networks. (**A**) Expression heatmaps of TF families, with the median FPKM value representing the expression level of each TF family and colors indicating the normalized FPKM values. (**B**) Gene networks regulated by TFs, with central nodes representing the TFs.

**Figure 5 plants-14-02549-f005:**
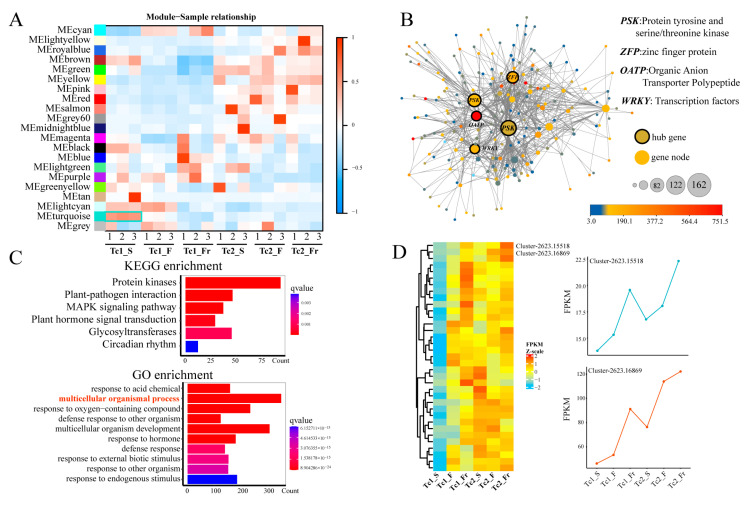
Identification of co-expression network modules. (**A**) Correlation between modules and sample traits. (**B**) The ME turquoise module network, with the dot colors representing the expression level of genes; each node represents a gene. The large nodes in the middle are considered hub genes. (**C**) GO and KEGG functional enrichment analysis of the ME turquoise module. (**D**) Expression heatmap of genes associated with the GO terms of multicellular organismal processes at different time points.

**Figure 6 plants-14-02549-f006:**
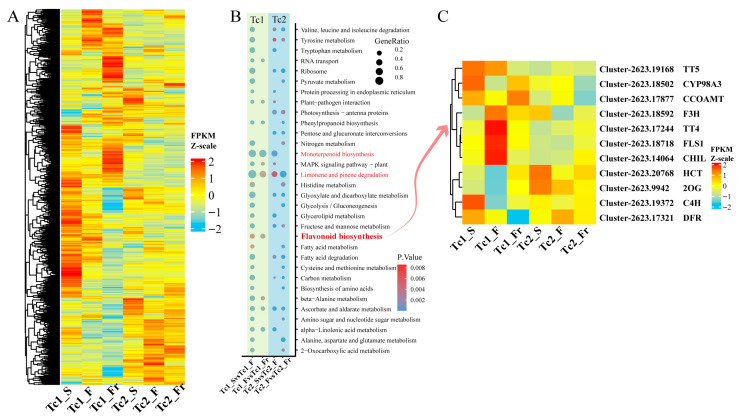
Functional annotation of differentially expressed genes. (**A**) Expression heatmap of differentially expressed genes at different time points; each row represents a gene. (**B**) KEGG functional annotation of differentially expressed genes at different time points. (**C**) Heatmap of flavonoid biosynthesis pathway-related genes.

**Figure 7 plants-14-02549-f007:**
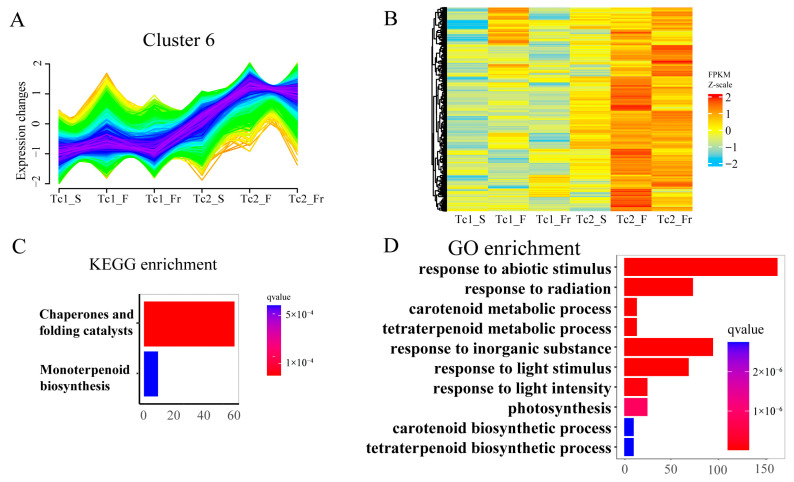
Cluster 6 gene set expression and functional annotation. (**A**) Trend of differentially expressed genes over time in Cluster 6. (**B**) Cluster 6 heatmap of differentially expressed genes across time points. (**C**) KEGG enrichment analysis of Cluster 6-related genes. (**D**) GO enrichment analysis of Cluster 6-related genes.

**Figure 8 plants-14-02549-f008:**
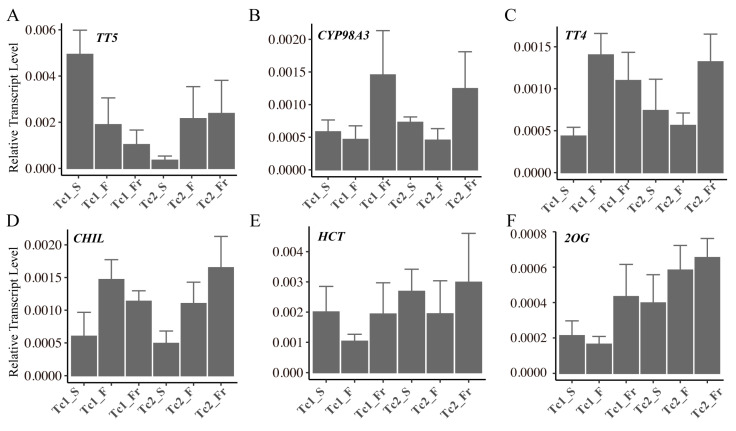
Relative expression levels of the relevant genes quantified using qRT-PCR. (**A**) *TT5*, (**B**) *CYP98A3*, (**C**) *TT4*, (**D**) *CHIL*, (**E**) *HCT*, (**F**) *2OG*. Each sample had four biological replicates, with the line graph representing the median of each sample.

**Table 1 plants-14-02549-t001:** Description of RNA sequences from the *T. chinense* growth cycle.

Sample	Clean_Reads	Clean_Bases	Q20	Q30	GC (%)
Tc1_S_1	20996764	6.30 G	97.05	92.19	47.98
Tc1_S_2	21020561	6.31 G	97.21	92.55	47.42
Tc1_S_3	21487470	6.45 G	96.81	91.62	47.46
Tc1_F_1	21400517	6.42 G	96.6	91.28	48.24
Tc1_F_2	21686559	6.51 G	97.26	92.59	48.16
Tc1_F_3	21351910	6.41 G	97.00	92.19	48.69
Tc1_Fr_1	20765996	6.23 G	96.73	91.62	47.94
Tc1_Fr_2	19902367	5.97 G	96.91	91.92	48.72
Tc1_Fr_3	21949511	6.58 G	97.00	92.15	48.68
Tc2_S_1	20824135	6.25 G	96.74	91.51	48.17
Tc2_S_2	20448756	6.13 G	96.98	92.02	48.20
Tc2_S_3	20575490	6.17 G	96.30	90.59	48.68
Tc2_F_1	20288826	6.09 G	97.06	92.2	47.48
Tc2_F_2	21461407	6.44 G	97.11	92.28	47.81
Tc2_F_3	21311564	6.39 G	97.14	92.40	48.92
Tc2_Fr_1	21006569	6.30 G	97.25	92.64	48.25
Tc2_Fr_2	21322398	6.40 G	97.03	92.14	48.41
Tc2_Fr_3	21074380	6.32 G	97.16	92.43	48.35

## Data Availability

The RNA-seq data from *T. chinense* that support the findings of this study are available in The National Center for Biotechnology Information Sequence Read Archive (SRA), reference number: PRJNA1171980.

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
