# Peer review of "Transcriptome Analysis of Developmental Gene Expression in *Thesium chinense* Turcz"

_plants, 2025, doi:10.3390/plants14162549_

Round 1

Reviewer 1 Report

Comments and Suggestions for Authors

Manuscript submitted to Plants, MDPI entitled:

Transcriptome Analysis of Developmental Gene Expression in Thesium chinense Turcz

 Sijia Liang1# , Qiongqiong Wang2,3# , Qin Han4 , Xinmin Zhang1 , Yiyuan Liu1 , Miaosheng Chen1 , Chengcai Zhang5 , 4 Zhaoyang Wang1 , Junxiao Li1 , Di Yu1 , Hao Zhan1 , Yubin Zhang1 , Sijia Liang1*, ZhongPing Xu

Dear authors,

Although the study has interesting potential, the manuscript has been submitted prematurely. Several sections in the current version need improvement and rewriting.

The manuscript requires language revisions throughout. For example, here is the corrected Abstract section.

Keywords: please do not use the exact wording as in the title.

Abstract: Thesium chinense Turcz. (T. chinense), A perennial herb in the Santalaceae family exhibits potent antibacterial and anti-inflammatory properties. Transcriptome sequencing was performed on one- and two-year-old T. chinense plants at the seedling, flowering, and fruiting stages, all collected from the exact location, using the Illumina HiSeq 4000 platform. A total of 58,706 unigenes were identified, including 1,656 transcription factors (TFs). Further analysis classified these TFs into eight categories, allowing for the inference of a typical TF regulatory network. Differential expression analysis revealed that the number of differentially expressed genes (DEGs) ranged from 2,000 to 22,500 across different developmental stages in first-year plants. At the same time, it varied between 1,000 and 2,000 in second-year plants. Moreover, the DEGs between one- and two-year-old T. chinense were primarily associated with sesquiterpene, triterpene, and terpene skeleton biosynthesis, as well as other metabolic pathways. Analysis of key genes involved in the biosynthesis of flavonoids, the major bioactive compounds in T. chinense, revealed that their predominant accumulation occurs during the first year of plant growth. This study offers valuable insights into the developmental biology of T. chinense and lays a foundation for future research on genes involved in flavonoid biosynthesis pathways.

Introduction

Please formulate a clear goal for this study.

Results

Overall, the Figures presentation needs improvement. It isn't easy to read several parts of them!

Figure 2 B, E

Figure 5 C, E

Figure 6: The entire set of results presented here is impossible to read and understand!

A complete botanical characterization of the plant species will be helpful (and is required) for readers not familiar with this medicinal plant species.

Discussion  

 His section needs some rewriting. Hardly on page in disunion is too short for this study.  The paragraph from the following lines is more suitable as an introduction to the Results, rather than the Discussion.

Lines 277-290

The actual discussion is missing here; the comparison with the literature data and the results obtained in this study needs to be adequately described. Comparison with other medicinal crops is also required. The role of flavonoids and the general knowledge database need to be discussed as well.

Authors also need to consider an explanation for the differences in gene expression across different years.  Needs to be correlated with various factors, such as climate, social, and agricultural practices, as well as the age and physiological status of the plants.

Materials and Methods

Please describe the growing conditions of plants in detail!

Conclusions

This section is completely missing in the current version of the manuscript! Please add it there and also explain how your results obtained in this study can be or will be used in the future!

Recommendation:

This manuscript requires improvement in several sections before it can be considered for publication.

17.7.2025

Comments on the Quality of English Language

The manuscript requires language revisions throughout.

Reviewer 2 Report

Comments and Suggestions for Authors

Reviewer 3 Report

Comments and Suggestions for Authors

Review of the manuscript: Transcriptome Analysis of Developmental Gene Expression in Thesium chinense Turcz

The research presented here addresses a timely and important topic: the molecular basis of flavonoid biosynthesis in a species of paramount medicinal importance that has so far received insufficient attention at the transcriptomic level. The authors have amassed a substantial data set and employed various sophisticated bioinformatics techniques. The validation of selected qRT-PCR results further enhances the reliability of the analysis.

Nevertheless, the manuscript requires substantial substantive and linguistic revisions before it can be considered for publication. Please find below a summary of the main points:

Introduction

The introduction to the manuscript provides a convincing rationale for conducting transcriptomic studies of Thesium chinense (T. chinense). The authors effectively identify knowledge gaps regarding the molecular mechanisms of synthesis of secondary metabolites, especially flavonoids, and present the phytopharmacological context of the plant. However, the section is encumbered by several substantive and stylistic issues requiring resolution. The content presented is somewhat exaggerated, contains repetitions, and overly general statements that are not reflected in the results analysed. Additionally, a clearly defined study objective and research hypothesis are lacking.
The authors accurately describe the substance's anti-inflammatory and antimicrobial properties and its use in traditional Chinese medicine. The information is well-supported by existing literature and is cross-sectional in nature. However, there is a lack of quantitative data to reinforce the importance of the statements. This could include levels of specific compounds or pharmacological activity.
The focus on the plant's biennial growth cycle and the challenges related to utilising wild populations is highly valuable. However, more detailed citations of data concerning population status and biomass are needed; at present, only general statements regarding low productivity and habitat degradation are provided.
The objective of the study was not clearly articulated.

Materials and methods

The section describes experiment design, RNA extraction protocols, library preparation, transcriptomic data analysis, and qRT-PCR validation. The authors provide essential information regarding the time and place of sample collection, the number of biological replicates, and bioinformatics tools. However, significant methodological details are absent, hindering the experiment's full replication. In some instances, the information provided is unclear or incomplete.
Please note that the collection year is missing from the material description. It has been said that the methodology for RNA isolation is not included. The phrase "standard extraction method" used is imprecise. The authors also omitted the details of the library preparation kit used and the sequencing parameters and technology. Additionally, the description lacks clarity regarding the type of RNA used for analysis, whether total RNA, the mRNA fraction, or rRNA depletion, which is crucial for understanding the findings.
Additionally, the description of the data analysis is lacking in significant detail. Please note that the information regarding the parameters of the programmes used is currently missing.
It should be noted that the description of the qRT-PCR reaction does not include primer sequences, and that there is no such table in the supplementary materials. Section 4.9 is redundant. This information should be included in the relevant analyses.

Results

The 'Results' section is extensive and includes descriptions of the RNA sequencing runs and the transcriptome assembly analyses, as well as the gene expression patterns and the regulation by transcription factors (TFs). It also covers the co-expression analysis (WGCNA) and the dynamics of flavonoid and terpenoid biosynthesis. The authors also present the validation of RNA-Seq data via qRT-PCR. While the section contains much relevant and valuable information, its structure, the presentation of results, and the quality of the language need significant improvement. The content is sometimes illegible or overly condensed, and lacks adequate interpretation. Several key figures and analytical details are also omitted or presented unclearly. Figures are, for the most part, barely legible. In Section 2.2 of BUSCO, 60% for the plant transcriptome is a relatively low value, but no comment was made. Data on the degree of transcript redundancy are missing. Why is the level of annotation in the KOG database so low compared to other databases? Section 2.3 lacks information on which genes are of particular interest — the analysis is limited to a numerical summary — and Section 2.4 lacks a list or table of the most relevant transcription factors (TFs) and their targets. Section 2.5 uses Clusters-2623.* as identifiers without explanation, which is incomprehensible to the reader. Section 2.6 does not indicate which genes changed the most (e.g., fold change), only general patterns. Figure 6 contains much information but is not referenced in the text. Figure 7 in Section 2.7 shows the results schematically; it would be better to compare RNA-Seq vs. qPCR plots.

Discussion

 Although the authors make several pertinent observations and link the results to practical contexts, such as breeding and extracting secondary metabolites, the section contains serious analytical and editorial shortcomings. The main issues are the superficiality of the interpretation, the failure to exploit the full potential of the data, the lack of critical self-reflection, and the presence of vague generalisations and tautologies. There is a lack of detailed analysis: which genes are responsible for the observed differences? Are there so-called 'hub genes' that differentiate between Tc1 and Tc2? The possible causes of downregulation in Tc2 were not commented on either. Is it the result of tissue ageing, hormonal differences, or environmental stresses? The specific effects of the missing genome were not mentioned, e.g, erroneous annotation of transcripts or false positives of DEGs. BUSCO = 60% is a low value,e, and no appropriate conclusions were drawn. There is also a lack of self-reflection: Is the data sufficient for network analysis? A performance model is also missing, e.g., a diagram showing how transcription factors (TFs) regulate flavonoid biosynthesis. The practical implications are not explored: could gene expression be manipulated to increase flavonoid production? 

Comments on the Quality of English Language

The language used in the manuscript is generally understandable, but it requires significant editing and proofreading. The writing style is overly technical in places and colloquial in others, resulting in stylistic inconsistency. Excessively long and complex sentences often hinder comprehension, as well as tautologies, repetitions, imprecise or vague wording, and syntactic and grammatical errors, including incorrect use of prepositions, awkward participle constructions, and incorrect word order.

Reviewer 4 Report

Comments and Suggestions for Authors

Please see attached

Round 2

Reviewer 1 Report

Comments and Suggestions for Authors

Hello,

The manuscript now has numerous positive corrections in several sections. 

Unfortunately, the authors did not take any action to improve the figures, which is essential.

Figure 5 is in very bad shape, as I was pointing out last time. This complex panel needs to be divided so that each section can be read and understood, especially the bottom part!

These are your results; they need to be adequately presented!

The same applies to Figure 6.

4.8.2025

Reviewer 2 Report

Comments and Suggestions for Authors

The manuscript is much improved. I'd like to thank the authors for addressing most of the comments.

There are a few minor points that still need to be addressed: 

  • Fonts in figures are still too small (for example, figure 1a, 1c, 2e) or the different text boxes go on top of each other (2a, 2c).
  • In Figure 2c, even though the R-squared between replicates is high, in many cases it is lower than between samples across sub-periods.  This should be pointed out and discussed clearly.
  • Figure 5d - It would be great to know which are the hub genes. Perhaps adding names to the figure or finding another way to convey the gene's identity could help readers understand which genes are more meaningful.
  • I couldn't find the new SI figures, so I couldn't check them. 
  • Also, I am unable to open Table s2. 

Reviewer 3 Report

Comments and Suggestions for Authors

I have no additional comments.

Author Response

Many thanks to the reviewer for their detailed guidance on this manuscripts and for their time and efforts in improving this manuscripts!